# Bleaching correction for DNA measurements in highly diluted solutions using confocal microscopy

**Lorenz Tim Sparrenberg**[1,2]*, **Benjamin Greiner**[2], **Harald Peter Mathis**[2]

**1** Institute of Biotechnology, RWTH Aachen University, Aachen, Germany, **2** Fraunhofer Institute of Applied Information Technology FIT, Sankt Augustin, Germany

* lorenz.sparrenberg@fit.fraunhofer.de

## Abstract

Determining the concentration of nucleic acids in biological samples precisely and reliably still is a challenge. In particular when only very small sample quantities are available for analysis, the established fluorescence-based methods give insufficient results. Photo-bleaching is seen as the main reason for this. In this paper we present a method to correct for the photobleaching effect. Using confocal microscopy with single molecule sensitivity, we derived calibration curves from DNA solutions with defined fragment length. We analyzed dilution series over a wide range of concentrations (1 pg/μl—1000 pg/μl) and measured their specific diffusion coefficients employing fluorescence correlation spectroscopy. Using this information, we corrected the measured fluorescence intensity of the calibration solutions for photobleaching effects. We evaluated our method by analyzing a series of DNA mixtures of varying composition. For fragments smaller than 1000 bp, our method allows to determine sample concentrations with high precision in very small sample quantities (< 2 μl with concentrations < 20 pg/μl). Once the technical parameters are determined and remain stable in an established process, our improved calibration method will make measuring molecular biological samples of unknown sequence composition more efficient, accurate and sample-saving than previous methods.

## Introduction

In molecular biology, precise knowledge of molecule concentrations, especially nucleic acid concentration, is required. Various methods, such as molecular cloning or sequencing involve nucleic acids and depend on precise concentration data [1, 2]. Also, the analysis of tissue samples or the examination of expression patterns of cells require exact concentration data on the amounts of extracted nucleic acid [3–5]. PCR-based methods can analyze nucleic acid samples consisting of only a few templates and are therefore widely used in molecular biology [5, 6]. However, these methods rely on sequence information from the sample. Thus, PCR-based methods cannot help in the analysis of nucleic acid mixtures of unknown sequences. Fluorescence measurements are highly promising for this task because of their extraordinary

**Data Availability Statement:** The data are held in a public repository. Link: https://zenodo.org/record/3908961.

**Funding:** The author(s) received no specific funding for this work.

**Competing interests:** The authors have declared that no competing interests exist.

sensitivity that even allows measuring single-molecule events [7]. A number of fluorescent dyes are available that bind sequence-independently to nucleic acids and thus enable reliable labeling [8, 9]. Additionally, fluorescent dyes for labeling are inexpensive and easy to handle. To consume as little of the valuable nucleic acid sample as possible, researchers prefer measuring in highly diluted solutions with microliter volumes [2]. Confocal fluorescence microscopy can meet these challenges and is therefore the ideal candidate for measuring mixtures of unknown nucleic acid sequences.

To obtain reliable results in fluorescence-based concentration determination, we have to consider various effects. In confocal measuring, soaring power intensities occur which may be of the order of several 100 kW/$cm^2$ may occur [10], which would expose the fluorophores to an enormous load. That is why the phenomenon of photobleaching, where molecules can permanently lose their fluorescence property due to irradiation with excitation light [11], is particularly problematic. During confocal measurements on freely diffusing molecules, a stationary equilibrium between the particle streams of bleached and fluorescent molecules is established in the excitation volume. Therefore, on average an apparently lower intensity, and thus also a lower concentration, of fluorophores in the solution is measured due to the bleaching effect. The extent to which a measurement is influenced by photobleaching depends on various factors, such as the irradiance of the excitation light, the photochemical properties of the fluorophore used, and the proportion of oxygen in the solvent [11]. The probability that a fluorophore is bleached is directly related to the duration of irradiation by the excitation light [12]. As a result, the bleaching rate occurring in the solution is a function of the molecular size, since larger molecules diffuse more slowly than smaller ones and thus remain longer in the excitation volume. A similar effect is valid for mixtures of nucleic acids with arbitrary fragment length distribution. The only difference is that here the mean fragment length and the mean diffusion constant are the characteristic properties of the mixture for photobleaching. As a consequence, nucleic acid mixtures of the same mass concentration but with different molarity have different mean diffusion constants, due to photobleaching.

We have derived a new calibration procedure to account for this effect and to correct the effect of photobleaching in fluorescence measurements. We use fluorescence correlation spectroscopy in a confocal setup to analyze the diffusion properties of DNA solutions. With this information, we can correct the fluorescence measurements for photobleaching to achieve highly accurate results.

## Theory

Fluorescence correlation spectroscopy (FCS) is a widely used technique to analyze the diffusion behavior of particles. The idea behind FCS is the correlation in time of thermodynamic fluctuations in concentration of diffusing particles in solution in equilibrium [13]. Eq 1 gives the autocorrelation function to calculate the FCS and corresponds to the correlation of a time series with itself shifted by time $\tau$:

$$G(\tau) = \frac{\langle \delta I(t) \cdot \delta I(t + \tau) \rangle_t}{\langle I(t) \rangle_t^2} \tag{1}$$

where $I(t)$ is the measured intensity at time $t$. $\langle \ldots \rangle_t$ denotes an average over time:

$$\langle I(t) \rangle_t = \frac{1}{T} \int_1^T I(t) \, \mathrm{d}t \tag{2}$$

The intensity fluctuation $\delta I(t)$ at a certain time is:

$$\delta I(t) = I(t) - \langle I(t) \rangle_t \tag{3}$$

From this, the autocorrelation function turns into

$$G(\tau) \quad = \frac{\langle \delta I(t) \cdot \delta I(t + \tau) \rangle_t}{\langle I(t) \rangle_t^2} = \frac{\langle I(t) \cdot I(t + \tau) \rangle_t}{\langle I(t) \rangle_t^2} - 1. \tag{4}$$

In this form, we can easily compute the autocorrelation from a given fluorescence trace. On the basis of physical considerations, a model for the autocorrelation can be derived. For this, the properties of the laser profile and molecular diffusion properties of the sample are considered. This model gives ground to approximate the detection efficiency of a diffusing particle excited by a single-mode laser in a confocal setup by a Gaussian profile [14]:

$$I(x, y, z) = I_0 \exp\left(-2\frac{(x^2 + y^2)}{r_0^2}\right) \exp\left(-2\frac{z^2}{z_0^2}\right) \tag{5}$$

where $x$, $y$ and $z$ are the coordinates of the observation volume. $z$ is the direction of the laser beam. $r_0$ is the radius of the observation volume and $(2z_0)$ is the effective length of the volume. For the Gaussian distribution of the laser profile, we can write the autocorrelation function for one freely diffusing particle species as [15]:

$$G(\tau) = \frac{1}{\langle N \rangle} \frac{1}{1 + \frac{\tau}{\tau_D}} \frac{1}{\sqrt{1 + \frac{r_0^2}{z_0^2} \frac{\tau}{\tau_D}}} \tag{6}$$

Here, $N$ is the average number of molecules in the detection volume. $\tau_D$ is the average diffusion time of particles in the observation volume giving the characteristic decay scale of fluorescence fluctuations [16]. We use Eq 6 to fit the experimental FCS data to get the diffusion time $\tau_D$ from the measurement. At $\tau = 0$, we obtain the mean number of diffusing particles in the volume $\langle N \rangle$ or equivalently the mean concentration $\langle C \rangle$:

$$G(0) = \frac{1}{\langle N \rangle} = \frac{1}{V_{eff} \langle C \rangle} \tag{7}$$

Where the effective observation volume is:

$$V_{eff} = \pi^{3/2} r_0^2 z_0 \tag{8}$$

The diffusion coefficient of the diffusing molecules in solution is:

$$D = \frac{r_0^2}{4\tau_D} \tag{9}$$

Standard implementations of FCS methods do not provide absolute values of diffusion coefficients, since they require information about the geometric shape of the detection volume, which is challenging to measure independently. Therefore, measurements of the diffusion coefficient are relative and require additional measurements of a reference substance, e. g. Alexa Fluor 488 with known diffusion coefficient and concentration to derive the information about the geometric shape [16].

It is important to note that with small shifting times further effects, such as triplet state effects (microseconds) [17] or photodiode afterpulsing (nano- to microseconds) [18] dominate the autocorrelation, so that Eq 6 needs to be adapted. Since we analyze here polymeric

molecules with large diffusion times ($\tau_D > 1 \times 10^{-3}$ s), the important changes of the autocorrelation function take place at relatively large shifting times and these effects can be neglected.

## Materials and methods

### Confocal setup

For our experiments we used a home-built fluorescence microscope in a confocal setup (Laser: Lasos LDM F series, 90 mW, 486 nm; Filters: Linus 1% neutral filter, Bright Line® fluorescence filter 535/40; Dichroic mirror: Linus 500 LP; Objective lens: Zeiss LD Plan-Neofluar 63x / 0.75 korr, $\infty$ / 0-1.5; Detector: Avalanche photo diode from Micro Photon Devices PDM series 100 μm). The pinhole has a diameter of 100 μm. An ALV correlator card processed the fluorescence signal. Fig 1 shows a schematic description of the confocal setup and the simplified measurement principle. We decided to use a 63x longdistance air microscope objective, because we wanted to keep the costs for the setup as low as possible and at the same time keep sample and measurement handling simple. By using an air objective instead of an immersion objective the detection efficiency decreases due to the lower NA. However, test measurements showed a sufficiently good S/N ratio for data evaluation (see Fig 2). For the estimation of the geometric shape of the detection volume, we conducted measurements with the fluorescence dye Alexa Fluor 488 (ThermoFisher Scientific), which has a known diffusion coefficient of 435 $\mu m^2\,s^{-1}$ [19].

### Measuring operation

For each fluorescence measurement, we placed a 2 μl drop on a cover slip and positioned it under the objective lens with the drop pointing away from the lens. Then we approached the measuring position along the optical axis, which was 50 μm in the drop volume. A pause of 30 s after each approach ensured that we were close to an equilibrium of bleached and unbleached

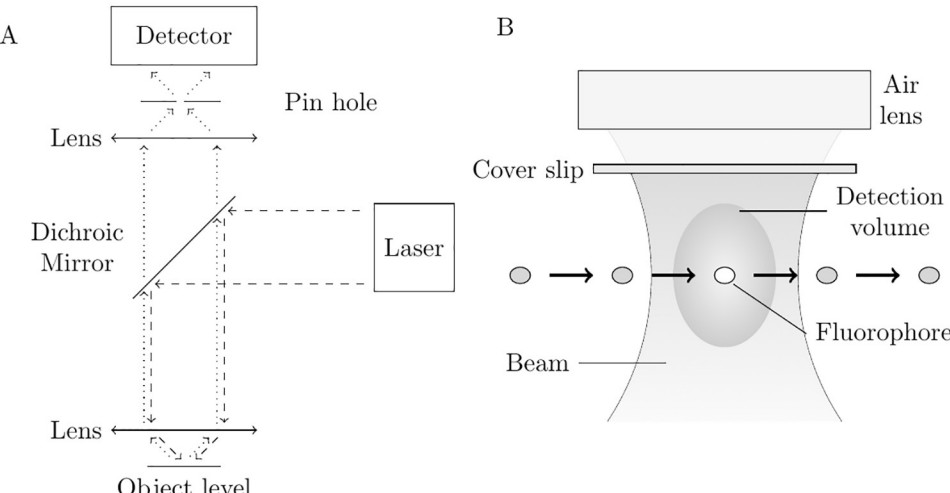

**Fig 1. Schematic of the experimental setup. A**: A laser emits light that is passed through a dichroic mirror and focused on the sample through an objective lens (dashed). Excited fluorophores in the probe start to emit photons which are collected by the objective lens. Because of their extended wavelength, these photons can pass the dichroic mirror. A detector counts these emitted photons (dotted). The pinhole improves the signal to noise ratio and assures that only photon events from the object level can reach the detector. **B**: The laser beam exits the objective lens and is focused behind the cover glass in the sample volume. A fluorophore enters by random walk the detection volume and begins to emit photons. The average time to pass the volume correlates with its diffusion coefficient.

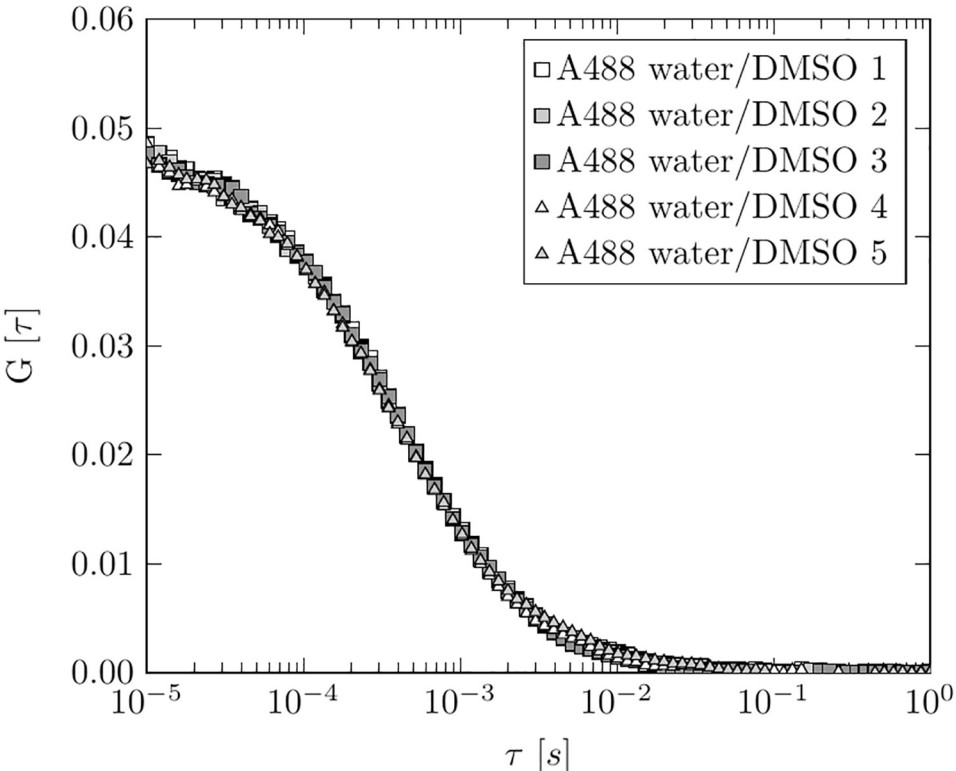

**Fig 2. Alexa Fluor 488 calibration measurements.** Fivefold measurement of Alexa Fluor 488 1 nM in 1 part 75% water/25% DMSO and 1 part TE-buffer. The FCS curves of the measurements show the characteristic curve of the autocorrelation function and can be fitted with Eq 6.

diffusing fluorophores leaving and entering the detection volume. The actual measurement took 30 s. During this time the ALV correlator card collected the data and calculated the auto-correlation of the measurement. The data were fitted to Eq 6 to get the characteristic diffusion time $\tau_D$ of each measurement. Each measurement was performed five times for different droplets. During all measurements the temperature was constant at 22˚.

## Sample preparation

For the calibration procedure, dilution series of dsDNA (NoLimits™, ThermoFisher Scientific) with fragment lengths of 50 bp, 200 bp, 500 bp, 1000 bp, 2000 bp, 3000 bp, 6000 bp and 10000 bp in 75% water/25% DMSO (vol/vol) were prepared. For labeling, the intercalator RiboGreen (Quant-iT™ RiboGreen™, ThermoFisher Scientific) was used. A solution of 3 μl RiboGreen stock in 997 μl TE-buffer was prepared and added to the DNA samples in a 1:1 ratio, yielding final mass concentrations of 1 pg/μl to 1000 pg/μl for the dilution series. After waiting one hour, fivefold measurements were taken for each concentration step. We want to point out that RiboGreen labels all types of nucleic acids unspecifically. Other intercalators such as Pico-Green label specifically double-stranded DNA, but have a significantly lower single molecule brightness and are therefore less suitable for FCS measurements in low concentration ranges.

The evaluation of the calibration method was carried out in a two-step process. In the first step, we set up DNA mixtures of known composition. The first set of mixtures consisted of 200 bp and 500 bp dsDNA with ratios of 1:1, 1:2, 1:3, 1:4 and 1:5 (wt/wt). The second set consisted of 50 bp and 1000 bp DNA with ratios of 1:1, 1:2, 1:3, 1:4, 1:5 and 1:6 (wt/wt). The

samples were again prepared in 75% water/25% DMSO (vol/vol) and labeled with the intercalator RiboGreen as described above, so that final concentrations of 20 pg/μl, 50 pg/μl, 100 pg/μl and 200 pg/μl were obtained. Again, we performed each measurement five times. The second step aimed to demonstrate the applicability of the method by determining the mass concentrations of 8 NGS libraries. The NGS libraries were exom libraries, which were fragmented by ultrasound. The libraries were adjusted to 1 ng/μl using the Qubit Flex Fluorometer (ThermoFisher Scientific). To demonstrate the capability of our method, we diluted the libraries with 75% water/25% DMSO (vol/vol) and labeled them with RiboGreen as described above, leading to theoretical concentrations of 10 pg/μl, 20 pg/μl and 50 pg/μl. We conducted our measurements five times and compared the results with the theoretical values. Because of the dilution steps and the small droplet volumes, we consumed only 0.8 μl of each library stock to determine the mass concentrations.

## Results and discussion

### Testing the confocal setup

To verify the correct functioning of our confocal set-up, measurements were performed with the fluorescent dye Alexa Fluor 488. Fig 2 shows an example of a fivefold measurement of Alexa Fluor 488 in 1 part 75% water/25% DMSO and 1 part TE-buffer. Thus, the measurements were performed using the same buffer composition as in the later DNA measurements. By fitting the autocorrelation data with Eq 6, the characteristic values of the investigated dye can be determined. For Alexa Fluor 488 in 1 part 75% water/25% DMSO and 1 part TE-buffer, we found a diffusion coefficient of D = 313 μm$^2$/s ± 11 μm$^2$/s. This is significantly lower than the 435 μm$^2$/s for Alexa 488 in pure water and reflects the higher viscosity of the buffer due to the addition of DMSO. Generally, the measurements prove the high reproducibility of the confocal setup. They also show that, although a longdistance air lens is used, enough photon events are collected to calculate a clean autocorrelation. This is important for the reliability of the subsequent DNA measurements.

### Diffusion properties of DNA

DNA measurements were carried out with the aim of establishing a highly precise mass concentration determination. To derive a calibration procedure to correct for photobleaching effects, we started by analyzing the diffusion properties of DNA solutions. For this purpose, we performed fluorescence measurements on DNA dilution series of defined fragment lengths. As an example, Fig 3 shows the autocorrelation curves of the FCS measurements exemplarily for the 50 bp dilution series. The black solid lines correspond to the fit of Eq 6 to the measurement data to get the specific diffusion time $\tau_D$ for each fragment length. We have considered only the data for shifting times greater than 1E-4 s in order to avoid problems such as triplet state effects or detector afterpulsing, which may occur at small shifting times in the range of nano- to microseconds. For increasing concentrations, the amplitudes of the graphs decrease, whereas $\tau_D$ remains constant for all graphs. The FCS results for the other DNA fragment sizes are similar to the results shown in Fig 3, with corresponding diffusion times $\tau_D$. Using Eq 9, we can calculate the mean diffusion coefficient for each DNA fragment size. Fig 4 shows the averaged diffusion coefficients for the different DNA fragments in a double-logarithmic representation. The diffusion coefficients seem to follow a power law. According to Zimm's model for flexible polymers in solution with excluded volume assumptions, for DNA solutions we would expect an exponent of around −0.60, because of the scaling law $D \propto M^{-0.60}$ [20], where $D$ is the diffusion coefficient and $M$ is the molecular weight. Actually, we found an exponent of −0.567 which is very close to previously reported values of −0.57 [21] and −0.571±0.014 [22]

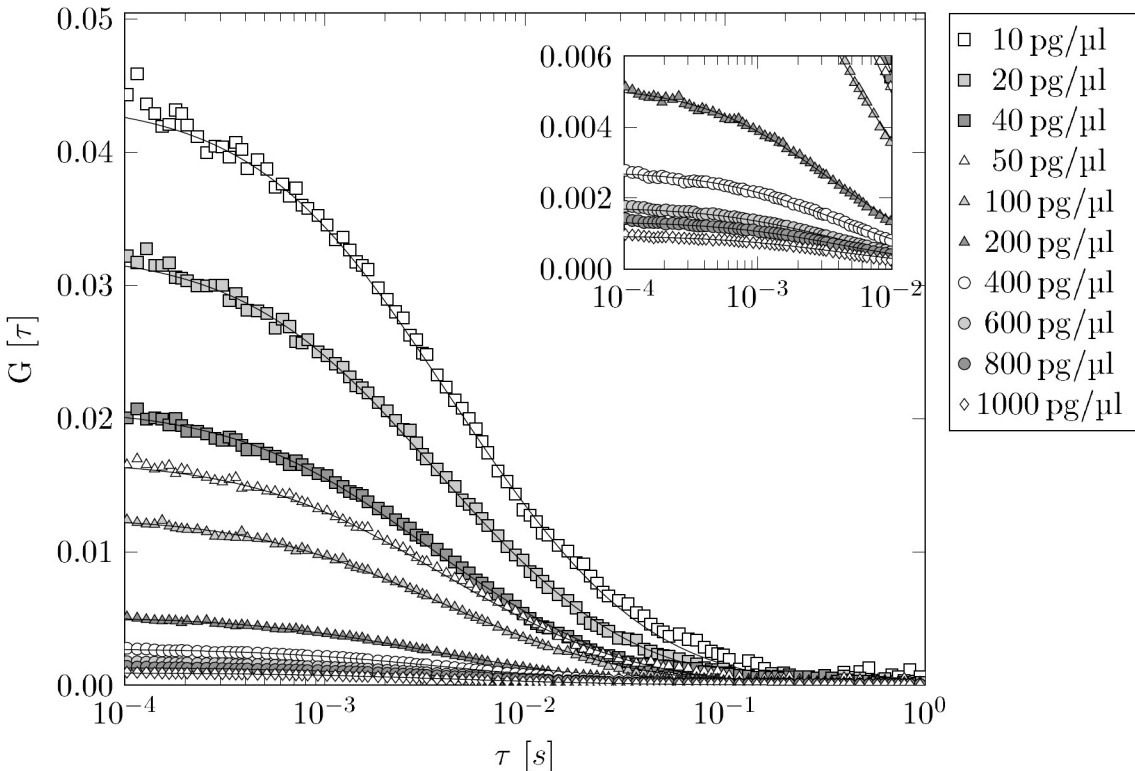

**Fig 3. FCS data of 50 bp DNA dilution series.** For clarity, we only show the median of the five individual measurements of each dilution step. The black solid lines correspond to the fit of Eq 6 to the data in order to get the specific $\tau_D$ values for each fragment length. For 1 pg/µl DNA solution (not shown) the FCS yields implausible results, because too few fluorescence events occur in the detection volume.

for dsDNA molecules in aqueous solution (see Fig 4). The exponent from the scaling law is independent of the viscosity and the ambient temperature during the experiments. Therefore, we can directly compare the exponents.

The results of the diffusion measurements and their comparison to literature show the high reliability of our measurements. During the measurements, we observed that with larger DNA fragments the fluctuations in the measured values tend to increase. Measurements on 20000 bp DNA fragments could no longer be evaluated meaningfully. However, even at fragment lengths of 2000 bp to 3000 bp, larger fluctuations became apparent. We would therefore advice to limit the measurement procedure to fragment lengths of up to 1000 bp in order to avoid additional sources of error. In the following sections, we have used the diffusion time $\tau_D$ instead of the diffusion coefficient $D$ to correct the bleaching effect.

## Bleaching correction of DNA measurements

The fluorescence intensity (more precisely: the fluorescence rate) will be used to determine the mass concentration. Therefore, we calculated the median of the fluorescence intensity over time for each dilution series of each fragment length. As an example, Fig 5 shows the fluorescence intensity for the dilution series of 50 bp DNA. The resulting graph is almost linear and we could fit the data with a straight line with sufficient accuracy. At high concentrations, however, the graph is best described by a polynomial of $2^{nd}$ order, since the quadratic term accounts for concentration-dependent effects such as quenching or volume exclusion. The

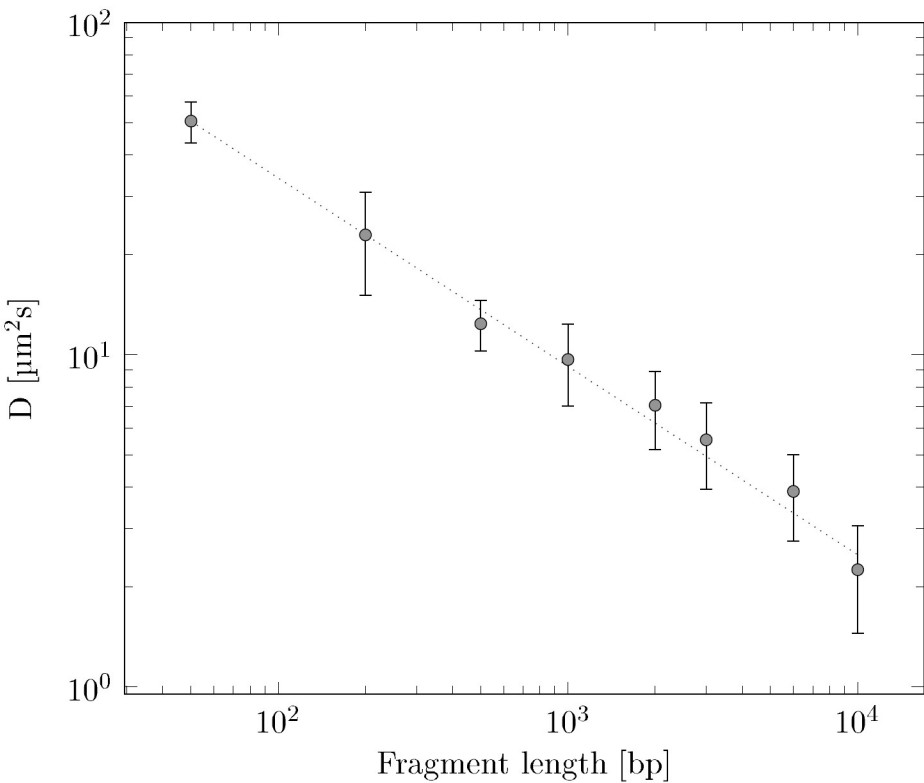

**Fig 4. Diffusion coefficients of DNA solutions of different fragment sizes.** The data were taken at 22˚. The circles correspond to the measurement data whereas the dotted line corresponds to the fitted model ($f(bp) = a \times bp^b$) with $a = 494.065$ and $b = -0.567$.

linear model in the S1 Appendix shows the derivation for bleaching correction with a linear fit. In the following, however, we will continue to work with a polynomial of the form

$$I = f(C) = a\,C + b\,C^2 + const \tag{10}$$

to describe the 50 bp dilution series. $I$ is the intensity and $C$ is the concentration of the analyzed solution. The $y$-axis intersection $const$ is set to the background noise we observed for each measurement. Fig 6 shows the intensities of all examined DNA dilution series. It is easy to see that for increasing fragment lengths the slopes of the intensities decreases. This effect is most probably due to photobleaching. For the determination of mass concentrations this poses a problem, since the mean fragment length is decisive for the correct calibration curve. To compensate for this effect, we rotate the fitted function $I = f(C)$ of the 50 bp dilution series around the $z$-axis intersection to describe the data of the other dilution series.

$$\vec{r'} = \begin{pmatrix} C' \\ I' \end{pmatrix} = R_\theta \vec{r} = \begin{pmatrix} \cos\theta & -\sin\theta \\ \sin\theta & \cos\theta \end{pmatrix} \begin{pmatrix} C \\ I \end{pmatrix} \tag{11}$$

where $C'$ and $I'$ are the measured concentration and intensity affected by photobleaching. By inserting $I = f(C)$ in Eq 11 and by translating it into the origin, we obtain for the expressions $C'$

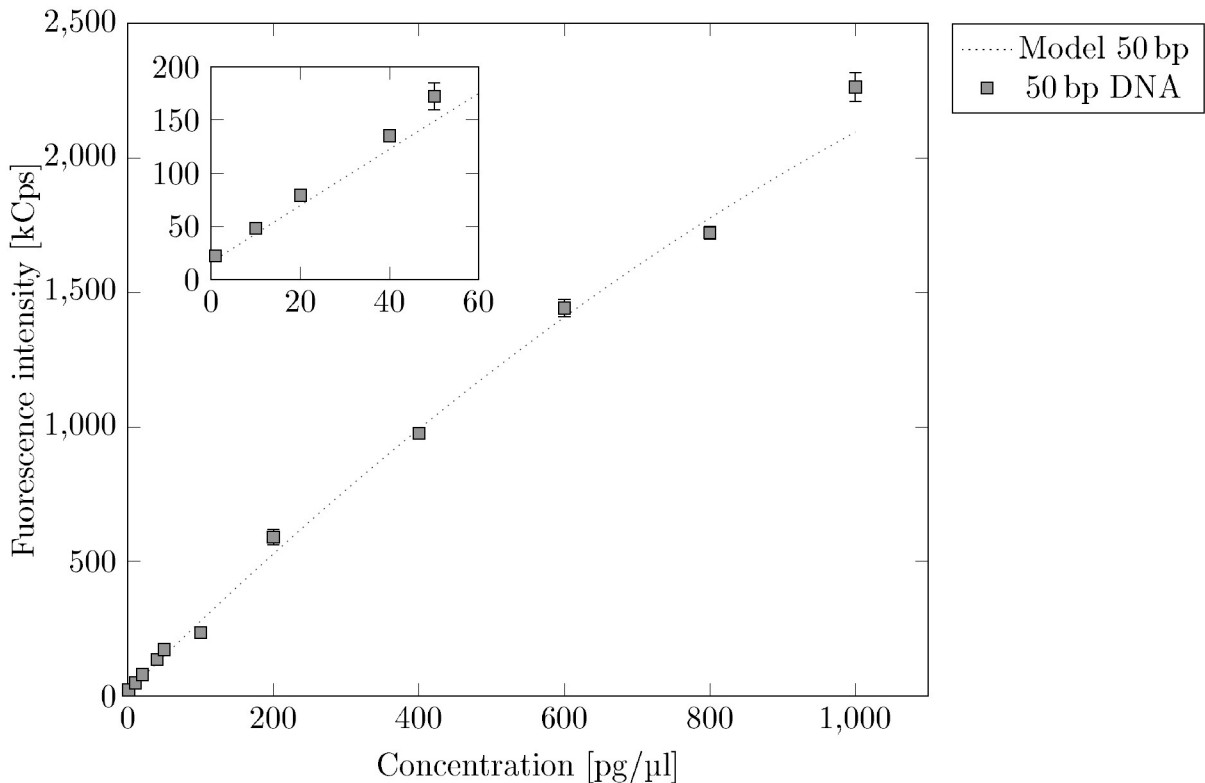

**Fig 5. Intensities of 50 bp DNA dilution series.** We used a polynomial with the shape $I = f(C) = aC + bC^2 + const$ to fit the data. With $const = 16.42$, we get $a = a_{cal} = 2.68$ and $b = -0.0006$.

and $I'$:

$$C'(\theta) = C\cos\theta - I\sin\theta = C\cos\theta - (a\,C + b\,C^2)\sin\theta \tag{12}$$

$$I'(\theta) = C\sin\theta + I\cos\theta = C\sin\theta + (a\,C + b\,C^2)\cos\theta \tag{13}$$

To express Eq 13 as a function of $C'$, we resolve Eq 12 to $C$:

$$C(\theta) = \begin{cases} \frac{1}{2b}\left(-a + \cot\theta \pm \csc\theta\sqrt{(\cos\theta - a\sin\theta)^2 - 4b\,C'\sin\theta}\right), & \text{for } [\theta \neq 0] \\ C', & \text{for } [\theta = 0] \end{cases} \tag{14}$$

In our case $\theta$ lies between $\left[-\arctan a < \theta < \frac{\pi}{2} - \arctan a\right]$ because only values in the first quadrant (positive concentrations and positive fluorescence rate) are reasonable. Furthermore, we only consider positive values for $a$ and concentration $C$, while $b$ has to be minimal and negative ($|b| \ll a$). Last but not least, only the negative term of Eq 14 is reasonable. Thus, we discard the positive term of Eq 14. By inserting Eq 14 into Eq 13 and by translating the expression

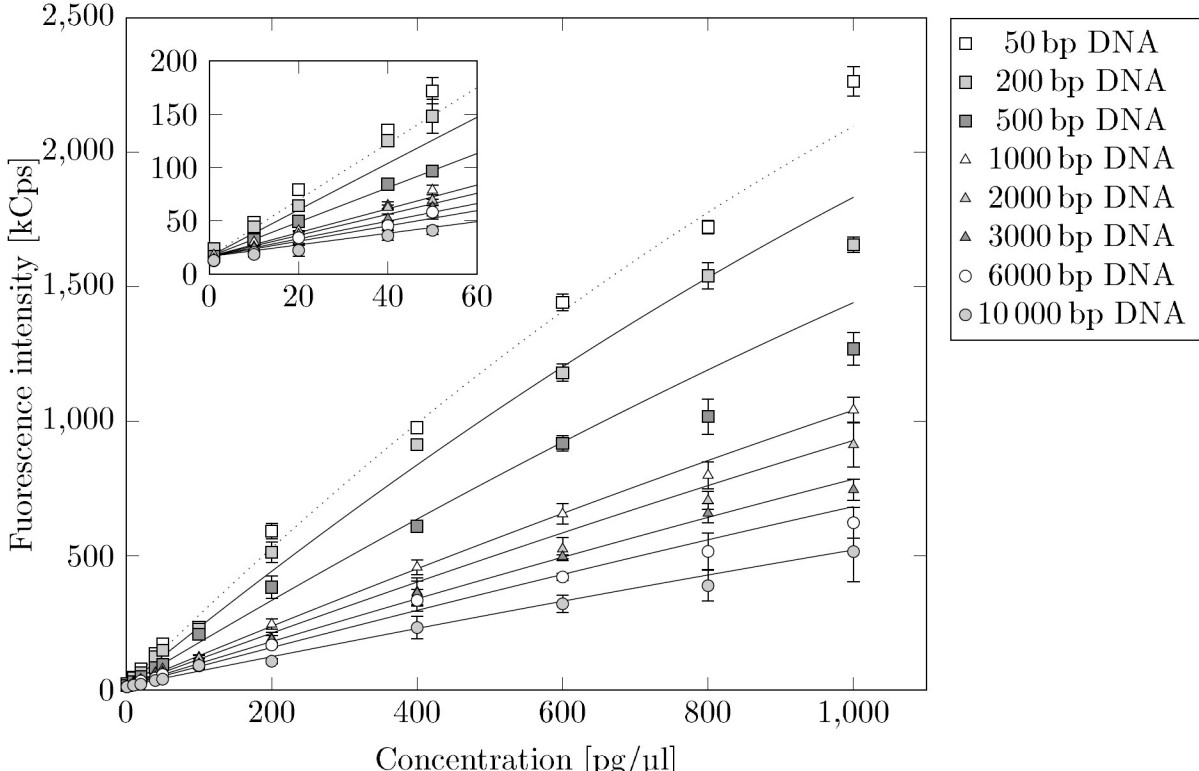

**Fig 6. Bleaching effect on dilution series of different DNA fragment sizes.** Dilution series of 50 bp, 200 bp, 500 bp, 1000 bp, 2000 bp, 3000 bp, 6000 bp and 10000 bp DNA fragments. We rotate the polynomial fit (dotted line) of 50 bp DNA clockwise to describe the other dilution series.

back to *const*, we get:

$$
\begin{aligned}
I'(C') \quad &= -\frac{1}{4b} \csc\theta \left( -\cos\theta + a\sin\theta + \sqrt{-4b\,C'\sin\theta + (\cos\theta - a\sin\theta)^2} \right) \\
&\times \left( a\cos\theta + \cos\theta\cot\theta + 2\sin\theta - \cot\theta\sqrt{-4b\,C'\sin\theta + (\cos\theta - a\sin\theta)^2} \right) + const
\end{aligned}
\tag{15}
$$

Eq 15 rotates Eq 10 around the *z*-axis intersection *const*. The validity of the approach is limited to functional areas where the rotated Eq 10 grows monotonously. For larger concentrations the quadratic term starts to dominate and the approach is no longer valid. Eq 15 is then fitted to the data of the remaining DNA dilution series (200 bp, 500 bp, 1000 bp, 2000 bp, 3000 bp, 6000 bp, 10000 bp) to get the rotation angle $\theta$ for each fragment length (see Fig 6). We are aware of the fact that we can fit each dilution series directly to a polynomial function without the detour via rotation. But the procedure using one calibration curve and rotating it to fit the data seems to be much more stable. The resulting rotation angle $\theta$ for each dilution series provides the corresponding slope $a_{sample}$. For this we rotate slope $a$ around the angle $\theta$.

$$
a_{sample}(\theta) = \frac{\sin\theta + a_{cal}\cos\theta}{\cos\theta - a_{cal}\sin\theta} = \frac{\tan\theta + a_{cal}}{1 - a_{cal}\tan\theta}
\tag{16}
$$

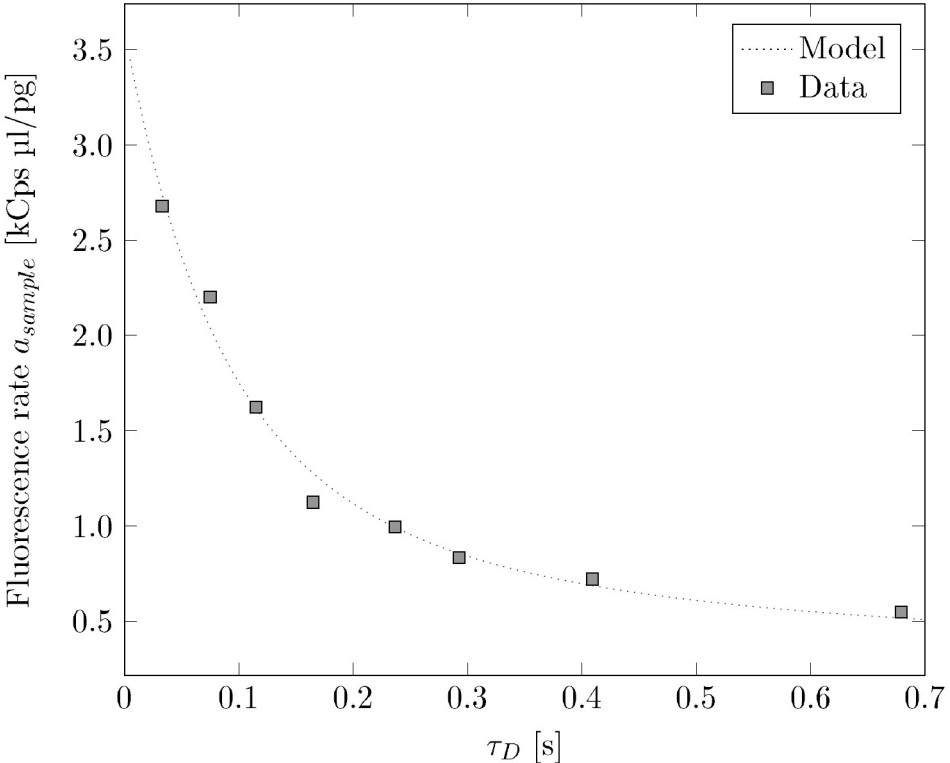

**Fig 7. Slopes of the bleached dilution series against the diffusion times.** The fluorescence rate as function of diffusion time for 50 bp, 200 bp, 500 bp, 1000 bp, 2000 bp, 3000 bp, 6000 bp and 10000 bp DNA fragments. Fitting Eq 18 to the data yields $k_{int}$ = 0.176, $k_{bl}$ = 19.016 and *const* = 0.257. The photobleaching effect affects the fluorescence rate which is hence lower for longer diffusion times (larger molecules).

Here, the slope $a_{cal}$ = $a$ comes from the 50 bp dilution series in Fig 5. Now, we plot the resulting slope of each rotated curve against its diffusion time. The result is shown in Fig 7. It can be clearly seen how the slope of the intensities of the dilution series decreases for increasing diffusion times and accordingly for increasing fragment lengths. To model the photobleaching as a function of $\tau_D$, we employ a probability-based approach:

$$N_f(\tau_D) = \frac{k_f}{k_{bl}}(1 - e^{-k_{bl}\,\tau_D}) \tag{17}$$

Here $N_f$ is the average number of fluorescence photons and $k_f$ and $k_{bl}$ are the fluorescence rate and bleaching rate, respectively [23]. By setting $k_{int} = \frac{k_f}{k_{bl}}$ and dividing the expression by $\tau_D$, we obtain the rate of fluorescent photons depending on the diffusion time. Finally, the introduction of a constant *const* is necessary to account for the fact that the fluorescence rate for long diffusion times can never be zero but is approaching a limiting value. Bringing all these considerations together, Eq 17 turns into:

$$f(\tau_D) = a_{sample} = \frac{k_{int}}{\tau_D}(1 - e^{-k_{bl}\,\tau_D}) + const \tag{18}$$

Fitting Eq 18 to the data yields: $k_{int}$ = 0.1759, $k_{bl}$ = 19.0164 and *const* = 0.2571 (see Fig 7).

## Determining mass concentrations of mixtures

In this section we show that knowledge of the increasing bleaching effect for large DNA fragments can be used to precisely determine the mass concentration of DNA mixtures. To determine the exact mass concentrations of a DNA sample of unknown composition, the following steps are performed:

1. Measuring the fluorescence intensity $I'$

2. Determining $\tau_D$ by calculation of the autocorrelation and fitting the data to Eq 6

3. Calculating $a_{sample}$ using Eq 18

4. Determining the angle of rotation $\theta$ via $a_{sample}$ and $a_{cal}$ from the calibration (50 bp DNA) using Eq 16

5. Calculating the corrected concentration $C'$ via Eq 15 and $I'$

The fluorescence intensity $I'$ is obtained directly from the measurements. The mean diffusion time $\tau_D$ for a sample is then determined from the measurements using the autocorrelation. Eq 18 gives the characteristic slope $a_{sample}$ for a given diffusion time $\tau_D$. Now, resolving Eq 16 to the angle of rotation $\theta$ yields the reverse function $\theta$:

$$\theta = \arctan\left(\frac{-a_{cal} + a_{sample}}{1 + a_{cal}\, a_{sample}}\right) \tag{19}$$

By inserting $a_{sample}$ from the step before and $a_{cal}$ from the 50 bp calibration measurement, we get the resulting rotation angle $\theta$ of the sample. To calculate the actual concentration of the DNA sample, we use the reverse function $C(\theta)$ of Eq 15:

$$
\begin{aligned}
C'(\theta) \quad =\ & \frac{1}{4}\Bigg( -\frac{2a\cos\theta)}{b} - \frac{2\sin\theta}{b} + 4\,const\,\tan\theta - 4\,I'\tan\theta - \frac{2a\sin\theta\tan\theta}{b} \\[6pt]
& - \frac{2\sin\theta\tan^2\theta}{b} - \frac{1}{b}\sqrt{2}\sin\theta \times \Bigg\{ -8b\,const\,\cot\theta\csc^5\theta + 8b\,I'\cot\theta\csc^5\theta \\[6pt]
& + \csc^6\theta + a^2\csc^6\theta - \cos(2\theta)\csc^6\theta + a^2\cos(2\theta)\csc^6\theta + 2a\csc^6\theta\sin(2\theta)\Bigg\}^{1/2} \\[6pt]
& \times \tan^2\theta \Bigg)
\end{aligned}
\tag{20}
$$

Inserting the calculated angle $\theta$ and the measured intensity $I'$ of the sample gives us the mass concentration of the DNA sample.

**Artificial DNA mixtures.** To evaluate the procedure, we used a set of 11 DNA mixtures, each in concentrations of 20 pg/µl, 50 pg/µl, 100 pg/µl and 200 pg/µl. We conducted each measurement five times and analyzed the fluorescence traces by means of FCS to determine the corresponding diffusion time $\tau_D$. Taking the median value of the five measurements, we calculated the corrected concentrations for each DNA mixture using the procedure described above. For comparison, we calculated the concentrations of the DNA mixtures according to the uncorrected standard procedure. For this purpose, we used the 50 bp dilution series (see Fig 5) as calibration standard and calculated the concentrations of the DNA mixtures on the basis of the fitted calibration measuring points. Fig 8 shows a comparison of the corrected results versus the uncorrected results. The data for all 11 mixtures are documented in S1

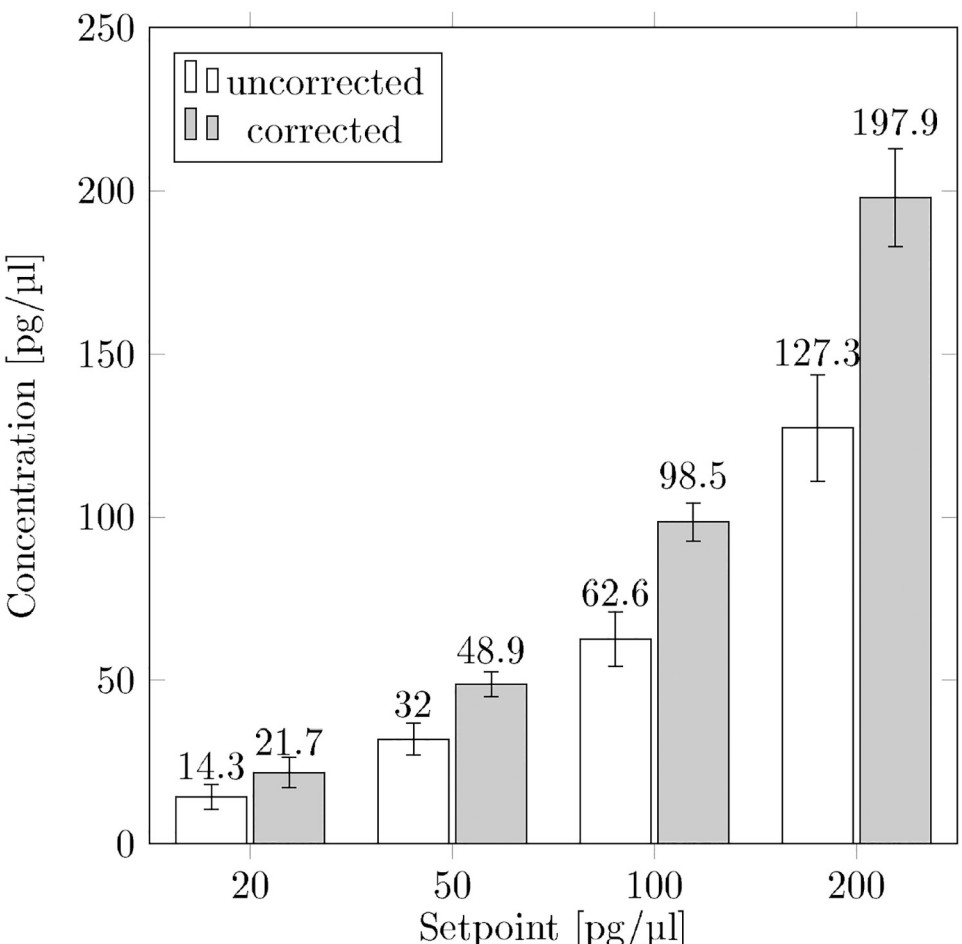

**Fig 8. Results of the determination of mass concentration.** The mass concentration of eleven DNA mixtures (2 μl drops) with setpoint concentrations of 20 pg/μl, 50 pg/μl, 100 pg/μl and 200 pg/μl are determined. Comparison of our calibration procedure to the uncorrected conventional procedure using the fluorescence intensity of 50 bp dilution series (see Fig 5) directly to determine the mass concentration.

Table in the S1 Appendix. For our procedure presented here, the deviation from the setpoint concentrations were below 2.3% for the 50 pg/μl, 100 pg/μl and 200 pg/μl samples. Even for the 20 pg/μl samples the deviation was below 8.6%. Without the bleaching correction, the derived mass concentrations were significantly underestimated compared to the setpoint values and showed larger standard deviations.

**Natural DNA mixtures.** In order to test the applicability to natural mixtures, we performed measurements on characterized NGS libraries. The libraries were diluted to 1:100, 1:50 and 1:20 of their original concentration and the mass concentration was determined. For this purpose, we used the calibration measurement of the 50 bp DNA dilution series on the one hand and the bleaching correction presented above on the other hand. The results are shown in Fig 9. The complete data of all mixtures are documented in S2 Table in the S1 Appendix. With the correction method presented here, empirical data are fairly close to the theoretical mass concentrations, whereas with the conventional method, mass concentrations are systematically underestimated. The mass concentrations determined from the NGS libraries via the bleaching correction have deviations of 1.2% to 4.7% percent from the expected concentrations. These are very small deviations for highly diluted solutions, especially since the stock

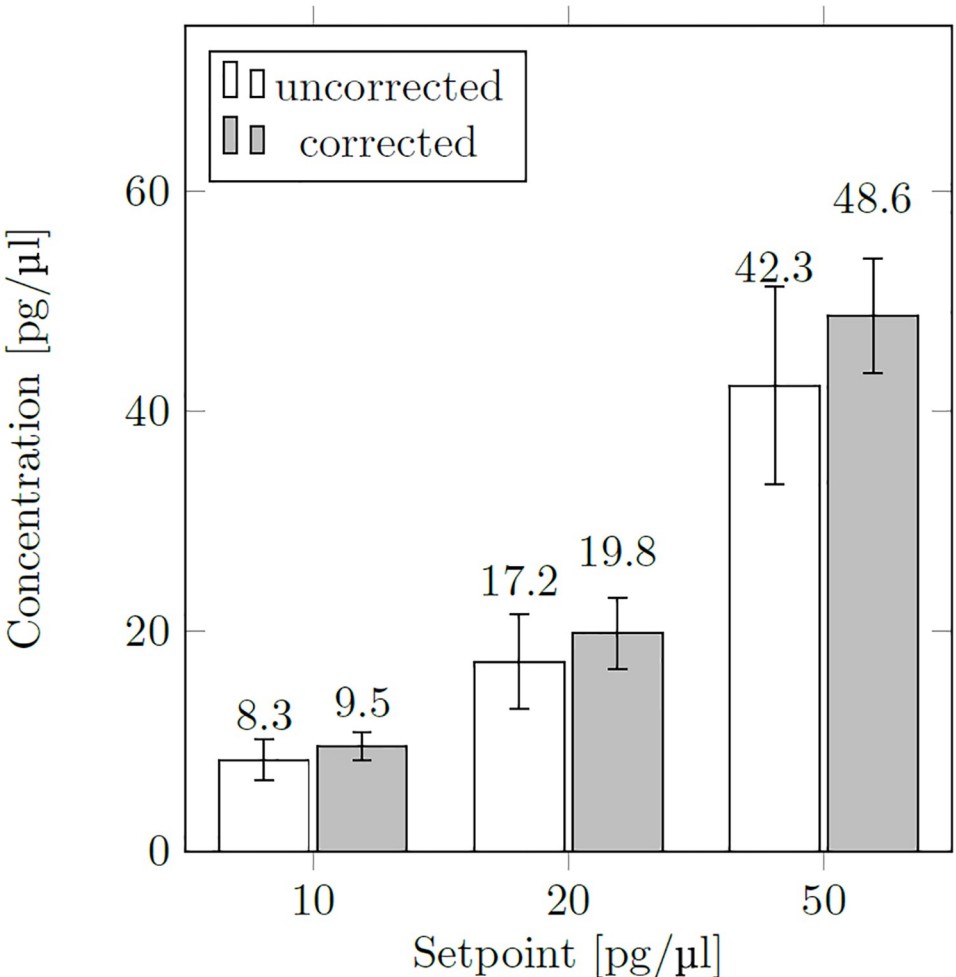

**Fig 9. Averaged mass concentrations of dilution steps of 8 NGS libraries (2 μl drops).** The concentration of the stock solutions was 1000 pg/μl. The dilution steps were 1:100, 1:50 and 1:20 yielding 10 pg/μl, 20 pg/μl and 50 pg/μl. Comparison of the new calibration procedure to the uncorrected conventional procedure to determine the mass concentration.

solutions of the libraries in our study were adjusted at significantly higher concentrations. Other methods using the Qubit or Nanodrop, require either significantly larger sample volumes or significantly higher concentrations of the initial solution to determine the mass concentration. For the analysis presented here, we only needed a total of 0.8 μl of the original stock. This was enough to prepare and examine the 1:100, 1:50 and 1:20 dilutions in fivefold preparations. In this way, as little as possible of the valuable sample is consumed; more sample can be used for other purposes. Thus, our method of correcting for photobleaching is of particular interest if only small sample quantities are available or for other reasons only highly diluted solutions can be investigated.

## Final considerations

The method presented here can determine very low concentrations of nucleic acid (10 pg/μl) in very small sample volumes (2 μl). However, various factors influence the calibration curve required to correct for bleaching: if they change, a new calibration is mandatory. In particular,

the slopes of the intensity curves depend on the laser power used. A low laser power leads to a less pronounced bleaching effect. At the same time, however, the measuring accuracy deteriorates due to a decreasing S/N ratio. This requires a good balancing of the effects to achieve the best results. Furthermore, the intercalators used for staining influence the bleaching correction. Due to different photo kinetics, the calibration curves need to be determined anew for each dye. The same applies to the buffers used. It is well known that ions and other buffer components have an influence on the brightness of the fluorescent dyes. Thus, the buffer used also affects the correction. Another important factor is the maximum fragment length of the samples under study. As we explained above, the method is particularly well suited for solutions with fragment lengths of $<1000 bp$. With larger fragments, there is significant variability in the measurements. Finally, we would like to point out that, besides photobleaching, the saturation of optical intensity is an important factor for fluorescence intensity. While the saturation of the fluorescence intensity only occurs at soaring laser intensities, significant parts of the molecules can already change into long-lasting triplet states at considerably lower power levels. Since our measurements were all taken at the same laser power, the relative deviation due to this effect is the same for all our measurements and can be neglected in the bleaching correction. Due to the complex calibration necessary, the method presented here is appropriate when only very small sample quantities are available for the analysis or when the parameters are fixed in an established process. Also, the fragments under study should be smaller than 1000 bp. We believe that, due to these restrictions, the method presented here is particularly well suited to quality control in NGS, where only very small sample quantities are available and the fragment length mixtures under study average about 200 bp to 400 bp.

## Conclusion

In this paper we presented a procedure to measure, with high accuracy, mass concentrations of DNA in highly diluted solutions. The challenge is to correct for the photobleaching effect which reduces the fluorescence rate of the sample. The larger the hydrodynamic radius of the sample the larger is the photobleaching effect. In a first step, we determined the diffusion properties of the sample by means of fluorescence correlation spectroscopy. Then, using the measured diffusion time data, the measured fluorescence intensity data are corrected for the bleaching effect. This allows a very precise determination of the mass concentration. For diluted NGS libraries, we determined an average of 9.51 pg/μl, 19.8 pg/μl and 48.6 pg/μl, results very close to the expected concentrations of 10 pg/μl, 20 pg/μl and 50 pg/μl, which we should obtain from the 1:100, 1:50 and 1:20 dilutions of the 1000 pg/μl stock solutions. These are remarkable results, considering that we conducted the measurements in tiny volumes of 2 μl and only needed 0.8 μl to prepare the three dilution steps for the fivefold measurements. And even measurements in 1 μl droplet volumes are feasible. This means that in the case of 10 pg/μl our method only needs a 10 pg sample of DNA to provide accurate results without the cost of expensive consumables. As it measures the average diffusion time for each sample, our method allows, in principle, to calculate the average DNA fragment size of the sample and thus the determination of the sample molarity. It is also thinkable to determine the degree of fragmentation of nucleic acids in a sample. This opens up interesting application fields in DNA and RNA extraction from rare samples such as tissue sections. Furthermore, this can help to avoid time-consuming and expensive examinations using capillary electrophoretic methods, for example. We believe that, as long as well-defined standard operating procedures can be followed, our improved calibration method will make measuring molecular biological samples of unknown sequence composition effortless, accurate and sample-saving when compared to previous methods.

## Supporting information

**S1 Appendix.**
(PDF)

## Acknowledgments

Besides the authors, several other members of the BioMOS group at Fraunhofer FIT were involved in this work. We thank them for their support.

## Author Contributions

**Conceptualization:** Lorenz Tim Sparrenberg, Benjamin Greiner.

**Data curation:** Benjamin Greiner.

**Formal analysis:** Benjamin Greiner.

**Investigation:** Lorenz Tim Sparrenberg.

**Methodology:** Benjamin Greiner.

**Project administration:** Harald Peter Mathis.

**Resources:** Harald Peter Mathis.

**Supervision:** Lorenz Tim Sparrenberg, Harald Peter Mathis.

**Visualization:** Lorenz Tim Sparrenberg, Benjamin Greiner.

**Writing – original draft:** Lorenz Tim Sparrenberg.

**Writing – review & editing:** Lorenz Tim Sparrenberg.

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
