## [Decision Letter · Decision Letter 0]

23 Apr 2020

PONE-D-20-09438

Bleaching correction for DNA-measurements in highly diluted solutions using confocal microscopy

PLOS ONE

Dear Mr Sparrenberg,

Thank you for submitting your manuscript to PLOS ONE. After careful consideration, we feel that it has merit but does not fully meet PLOS ONE’s publication criteria as it currently stands. Therefore, we invite you to submit a revised version of the manuscript that addresses the points raised during the review process.

We would appreciate receiving your revised manuscript by Jun 07 2020 11:59PM. To enhance the reproducibility of your results, we recommend that if applicable you deposit your laboratory protocols in protocols.io, where a protocol can be assigned its own identifier (DOI) such that it can be cited independently in the future. For instructions see: http://journals.plos.org/plosone/s/submission-guidelines#loc-laboratory-protocols

We look forward to receiving your revised manuscript.

Kind regards,

Joseph Banoub, Ph,D., D. Sc.

Academic Editor

PLOS ONE

Reviewers' comments:

Reviewer's Responses to Questions

**Comments to the Author**

1. Is the manuscript technically sound, and do the data support the conclusions?

Reviewer #1: Partly

Reviewer #2: Partly

2. Has the statistical analysis been performed appropriately and rigorously? 

Reviewer #1: I Don't Know

Reviewer #2: N/A

3. Have the authors made all data underlying the findings in their manuscript fully available?

Reviewer #1: No

Reviewer #2: Yes

4. Is the manuscript presented in an intelligible fashion and written in standard English?

Reviewer #1: Yes

Reviewer #2: No

5. Review Comments to the Author

Reviewer #1: Dear Author, I enjoyed reading about your novel approach using a FCS based diffusion time method to determine DNA fragment sizes. There are a few items of concern that I have listed below.

To make this study more believable it should have more controls. The experiments were conducted using clean, known size DNA fragments. It is common to compare the results of known samples to actual DNA samples to demonstrate applicability. Did you include any DMSO/Alexa488 only samples?

It would be good to describe the benefit of your method over others or how this system could be used specifically in research. For example, you do not mention spectroscopy (i.e. nanodrop) as a comparative way to quantify small samples of DNA.

The microscope setup includes a Zeiss LD Plan-Neofluor 63x, 0.75NA objective. This objective would have a large depth of field, air to liquid mismatch, and is not recommended by Zeiss for high resolution imaging. The diagram is also misleading because this objective is a long working distance objective.

Reviewer #2: In the manuscript titled “Bleaching correction for DNA-measurements in highly diluted solutions using confocal microscopy”, Sparrenberg et al. developed a confocal microscopy method to examine DNA concentrations at concentrations of less than 20 pg/μl. Specific comments are as follows:

1) The DNA being examined is pretty precise and it can represent sizes that are consistent with plasmids or short digested fragments, but there is no representation of larger biologically relevant sequences (such as a BAC, YAC, or genomic DNA). Please assess at least genomic DNA at comparably low concentrations using your system. If this cannot be accurately assessed, explain the limitation in the Discussion.

2) DNA samples that are being used for molecular biology manipulation typically are diluted in a variety of buffers, ranging from water (as in the manuscript, together with DMSO), NaCl, Tris-EDTA, among others. Please examine if these matrices affect measurements in your system.

3) Language needs to be improved, for grammar and spelling.

6. PLOS authors have the option to publish the peer review history of their article (what does this mean?). If published, this will include your full peer review and any attached files.

Reviewer #1: No

Reviewer #2: No

---

## [Author Response · Author response to Decision Letter 0]

29 Jun 2020

Dear Sir or Madame,

Thank you for taking the time and effort to critically review our paper draft. In the next few lines we would like to address the points of criticism you have raised. 

Reviewer #1: Dear Author, I enjoyed reading about your novel approach using a FCS based diffusion time method to determine DNA fragment sizes. There are a few items of concern that I have listed below.

1) To make this study more believable it should have more controls. The experiments were conducted using clean, known size DNA fragments. It is common to compare the results of known samples to actual DNA samples to demonstrate applicability. Did you include any DMSO/Alexa488 only samples?

Response: We have extended our work by measurements on natural mixtures (fragmented exom libraries for next generation sequencing). These indeed round off the results and have been missing so far.

Additionally, we have carried out evaluation measurements of the self-built measuring system in advance. Since there are no references to this in the script so far, we have added evaluation measurements of Alexa 488 in water/DMSO at the beginning of the result/discussion section.

2) It would be good to describe the benefit of your method over others or how this system could be used specifically in research. For example, you do not mention spectroscopy (i.e. nanodrop) as a comparative way to quantify small samples of DNA.

Response: Our method is particularly suitable for procedures where only small sample volumes are available in highly diluted amounts. At the same time, the DNA fragments examined should be < 1000bp. Since changes in the buffers or dyes used also require recalibration, the method should be used wherever solid SOPs exist. We therefore think that our method is particularly suitable for quality control in next generation sequencing. We have extended the discussion in the paper accordingly.

Other comparative spectroscopic methods used for mass determination of nucleic acids require either larger sample volumes (Qubit, ~20µl for 10pg/µl initial concentration) or significantly higher initial concentrations (Nanodrop, > 1ng/µl in 1-2µl sample volume). 

3) The microscope setup includes a Zeiss LD Plan-Neofluar 63x, 0.75NA objective. This objective would have a large depth of field, air to liquid mismatch, and is not recommended by Zeiss for high resolution imaging. The diagram is also misleading because this objective is a long working distance objective.

Response: We decided to use an LD lens because it offers some advantages. We are planning to integrate our measuring method into an automated measuring environment. A so-called autofocus will scan the cover glass of the sample in z-direction and find the suitable measuring plane. This procedure is technically much easier to implement with an air lens than with an immersion lens. We are aware that an air lens also has some disadvantages. We would like to go into these briefly. 

- Spherical aberration due to air-glass liquid phase boundaries: We can alleviate this effect by the fact that the LD lens used has a correction ring. The spherical aberration can be minimized by a suitable combination of cover glass correction and immersion depth in the measuring matrix.

- Smaller aperture angle of the lens due to smaller NA: Due to the smaller NA (0.75 instead of e.g. 1.4 for an 63x oil immersion lens) we lose some photon events during detection. However, this is acceptable. We have performed both comparison measurements between a 63x immersion objective with 1.4 NA and the 63x LD objective used on Alexa 488 solution (see Fig 1 in the attached pdf 'Response to Reviewers.pdf'). The results are somewhat less favorable for the LD lens, but acceptable. Most importantly, a clean autocorrelation can be calculated.

Our evaluation measurements have thus shown that the LD objective provides reliable results for our application. Finally, we adjusted the schematic of our setup to clarify that we are working with an air lens. 

Reviewer #2: In the manuscript titled “Bleaching correction for DNA-measurements in highly diluted solutions using confocal microscopy”, Sparrenberg et al. developed a confocal microscopy method to examine DNA concentrations at concentrations of less than 20 pg/μl. Specific comments are as follows:

1) The DNA being examined is pretty precise and it can represent sizes that are consistent with plasmids or short digested fragments, but there is no representation of larger biologically relevant sequences (such as a BAC, YAC, or genomic DNA). Please assess at least genomic DNA at comparably low concentrations using your system. If this cannot be accurately assessed, explain the limitation in the Discussion.

Response: The presented method is only suitable for fragment lengths of < 10000 bp. But already from about 2000 bp on large uncertainties in the measurements occur. This is mainly due to the expansion of the detection volume. Large DNA fragments exceed the detection volume in their expansion, which causes the correlation to become increasingly erroneous. We therefore recommend the procedure for fragment lengths < 1000 bp. These can be found, for example, in the characterization of DNA libraries for DNA sequencing. We have added a corresponding paragraph to the discussion.

2) DNA samples that are being used for molecular biology manipulation typically are diluted in a variety of buffers, ranging from water (as in the manuscript, together with DMSO), NaCl, Tris-EDTA, among others. Please examine if these matrices affect measurements in your system.

Response: The method can also be used for DNA in other solution buffers. However, this requires new calibration measurements for each buffer. It is known that ions have an influence on the brightness of intercalators like RiboGreen or PicoGreen. Since the method works with the fluorescence rates of the solutions, the influence of ions is an important factor. Also changes in the viscosity of the buffers (e.g. by the DMSO used) have an influence on the diffusion times of the molecules. Because the diffusion time is needed to correct the calibration curves, the viscosity of the buffer has an important influence on the correction procedure. We have extended the discussion by a section where we discuss these influencing variables on the correction procedure.

3) Language needs to be improved, for grammar and spelling.

Response: We had a proofreader correcting the text of the script. We hope that the text now meets the requirements for a publication in PLOS ONE.

I hope that we could clarify all open points. If you have any further questions, I will of course be at your disposal. We have deposited the data at https://zenodo.org (DOI: 10.5281/zenodo.3908961).

Thank you very much in advance.

Best regards,

Lorenz Sparrenberg

---

## [Editor Report · Decision Letter 1]

6 Jul 2020

Bleaching correction for DNA measurements in highly diluted solutions using confocal microscopy

PONE-D-20-09438R1

Dear Dr. Sparrenberg,

We’re pleased to inform you that your manuscript has been judged scientifically suitable for publication and will be formally accepted for publication once it meets all outstanding technical requirements.

Kind regards,

Joseph Banoub, Ph,D., D. Sc.

Academic Editor

PLOS ONE
---

## [Editor Report · Acceptance letter]

8 Jul 2020

PONE-D-20-09438R1 

Bleaching correction for DNA measurements in highly diluted solutions using confocal microscopy 

Dear Dr. Sparrenberg:

I'm pleased to inform you that your manuscript has been deemed suitable for publication in PLOS ONE. Congratulations! Your manuscript is now with our production department. 

Kind regards, 

on behalf of

Dr. Joseph Banoub 

Academic Editor

PLOS ONE